# Tracking the Activation of Heat Shock Signaling in Cellular Protection and Damage

**DOI:** 10.3390/cells11091561

**Published:** 2022-05-05

**Authors:** Shisui Torii, Pasko Rakic

**Affiliations:** Department of Neuroscience, School of Medicine, Yale University, New Haven, CT 06510, USA; shisuitorii@gmail.com

**Keywords:** Heat Shock signaling, detection of injury, epigenetics, Heat Shock Factor 1, Heat Shock Protein 70, reporter mice, genetic recombination, transcriptional bursting, heterogeneity, transcriptional amplification

## Abstract

Heat Shock (HS) signaling is activated in response to various types of cellular stress. This activation serves to protect cells from immediate threats in the surrounding environment. However, activation of HS signaling occurs in a heterogeneous manner within each cell population and can alter the epigenetic state of the cell, ultimately leading to long-term abnormalities in body function. Here, we summarize recent research findings obtained using molecular and genetic tools to track cells where HS signaling is activated. We then discuss the potential further applications of these tools, their limitations, and the necessary caveats in interpreting data obtained with these tools.

## 1. Heat Shock Factor 1 Is a Primary Controller of the Transcription of Heat Shock Protein Genes

Proteostasis is a function that keeps protein homeostasis in the cell. Upon exposure to cellular stress, the synthesis of Heat Shock Proteins (HSPs), which play key roles as molecular chaperones to prevent protein misfolding and aggregation, is significantly increased [1]. This is one of the essential molecular processes controlling proteostasis [2]. Transcriptional activation of *HSP* genes is processed by Heat Shock Factors (HSFs). Among those, HSF1 is a primary transcriptional factor in the stress response in vertebrates [3,4]. Using knockout mice and cellular models, researchers have demonstrated that HSF1 controls the transcription of *HSP* genes to maintain cellular integrity under exposure to various types of stress as well as to develop thermotolerance [5]. HSF1 is constitutively expressed in most tissue and cell types but remains inactive without cellular stress. Exposure to heat or other stressors changes the conformation of HSF1, causing it to bind DNA for transcription of *HSP* genes. This activation process of HSF1 is intricately regulated through multiple protein–protein interactions, trimerization, and subcellular localization [6].

## 2. Heat Shock Reporters Unravel the Molecular Mechanisms of Heat Shock Signaling Activation

Using the characteristics of HSF1 that primarily mediates *HSP* transcription, various reporter systems have been developed to detect HS signaling activation. Those HS reporter systems have been used to reveal the molecular players and interactions in HS signaling. For example, Rallu et al. [7] used a transgenic mouse line harboring a luciferase reporter gene under the control of the *Hsp70* promoter. Both quantitatively and qualitatively, the luciferase activity was far from parallel to the DNA-binding activity of HSF2, another member of the HSF family of transcription factors, during embryogenesis, suggesting that HSF2 is not involved in the regulation of *Hsp70* transcription in mouse embryonic development. Similarly, in yeast, a luciferase-based HS reporter was used to reveal the functions of two molecular chaperones, Ssa and Ssb (*Hsp70* homologs), in regulating HSF activity in both unstressed and heat-shocked cells [8]. Another example can be found in work by Feder et al. [9]. They used the HSF1 binding element, which is highly conserved from yeast to humans and present in the promoters of *HSPs* and other HSF1 target genes, to drive the Yellow Fluorescence Protein (YFP) encoding gene in yeast cells, and revealed that the rapid change in the interaction between Sis1 (an Hsp40 homolog) and other chaperone proteins was a prerequisite for Hsf1-mediated activation of gene transcription upon heat shock. 

## 3. Heat Shock Reporters Detect the Activation of Heat Shock Signaling upon Various Types of Cellular Stress and Injury

As described in the previous section, HS reporters using luciferase or fluorescent proteins have helped to elucidate the molecular mechanisms of HS signaling under normal and stressed conditions. Another major application of the HS reporter system is defining and monitoring tissues and cells that respond to various cellular stress and injury to maintain proteostasis in vivo and in vitro.

By leveraging noninvasive whole-body bioluminescence imaging with transgenic mice carrying a luciferase reporter gene under the control of the *Hsp70* promoter, activation of HS signaling by high temperatures in the laser-exposed skin area was confirmed [10]. The transgenic mice generated by another group harbor an *Hsp70* promoter-driven luciferase/green fluorescent protein (GFP) dual reporter. This mouse line was also shown to be useful for evaluating cellular stress and survival, collateral damage, and wound healing due to tissue injury, such as laser ablation of skin [11]. Far-red protein mPlum also was used to generate HS reporter transgenic mice [12]; these animals generated by de la Rosa et al. allow low-noise live imaging of the post-ischemic brain through a cranial window. 

Importantly, a tandem repeat of the HSF1 binding element, each of which consists of 25 nucleotides, was shown to drive HS reporter expression to a degree similar to that driven by the ~700 bp *Hsp70* promoter in various cell lines [13]. The same study also showed that brief exposure to a high temperature (50 °C for a few minutes) to recapitulate burns induces reporter expression comparable to the standard heat shock levels commonly used in research (39–43 °C for 1–2 h). Another study compared the temporal dynamics of reporter expression with those of *Hsp70* mRNA expression in mice using luciferase and mPlum under the control of the *Hsp70* promoter as reporters. The peak of luciferase activity was delayed by 3 h from the peak of *Hsp70* mRNA production, while the peak of the mPlum protein was delayed even more [14], indicating a limitation of reporter systems in terms of time resolution.

In *Caenorhabditis*
*elegans*, the expression of small *hsp*s is also controlled by HSF1, and thus a reporter transgene in this species was also used to observe the effect of various stresses, including oxidative stress and accumulation of human β amyloid peptides on cells, to investigate the molecular mechanisms of stress response conserved across species [15]. 

A study using a transgenic HS reporter system in a mouse model of spinal cord injury demonstrated electrophysiological changes in reporter-positive neurons 4 weeks after the injury [16], suggesting the applicability of HS reporter to trace specific cell populations that exhibit long-term, potentially epigenetic, changes caused by cellular stress. Another study using the same HS reporter system demonstrated that patterns of HS signaling activation in neurons and progenitors in the brain vary between embryos exposed to different environmental insults such as X-ray and ethanol [17]. The reporter was not expressed by other types of stressors such as nicotine and valproic acid. These results support that different stressors have specific effects on brain development through distinct molecular mechanisms. Using those reporter mice, Ishii et al. also found mosaic activation of HSF1 in the brain of mice prenatally exposed to ethanol [18]. This is consistent with the report that variable levels of *HSP70* mRNAs are observed among human-iPS-cell-derived neural progenitor cells exposed to oxidative stress in the same culture dish [19]. 

## 4. Generation of Permanent Tracing Systems for Cells That Activate Heat Shock Signaling

Prenatal exposure to the same or similar doses of harmful agents such as ethanol and heavy metals can have highly variable and unpredictable negative effects on the brain from fetal life through adulthood. The severity also varies. A group of researchers previously demonstrated that HSF1 is sporadically activated when the fetal brain is exposed to various harmful chemicals [17,19]. This activation of HS signaling protects brain cells from cell death upon exposure to these chemicals, presumably by maintaining proteostasis. In addition, they showed that excessive activation of HSF1 in a subpopulation of cortical cells detected by the HS reporter expression disrupts their normal developmental processes, such as migration [18]. However, it was unknown whether HS signaling activated by those stressors in early life is associated with later disease manifestation.

The group subsequently generated a new HS reporter system by which the cells that were once activated HS signaling in response to cellular stress or injury can be permanently labeled after the HSF1 activation diminishes. In this system, the driver construct consists of the *HSP70*-promoter-driven-flippase recombinase gene (FLPo), and the reporter construct consists of the FRT-stop-FRT-RFP so that the cells in which HSF1 is activated will express red fluorescent protein (RFP) permanently (Figure 1a). This system was shown to work in both an in utero electroporation-mediated gene transfer approach and in a transgenic mouse approach (*HSP70* promoter-FLPo mice are crossed with FRT-stop-FRT-RFP mice) [17,18,20,21]. 

Using this system, they found a random distribution of RFP^+^ neurons and glial cells in the brain in postnatal mice that were prenatally exposed to ethanol or sodium arsenite [17,20,21]. In addition, abnormal excitability was observed in those RFP^+^ cortical neurons but not in RFP^-^ cortical neurons in the same brain [20,21]. These findings indicate that the impacts of prenatal stress exposure on neural progenitor cells can lead to functional abnormalities in their daughter cells in the postnatal brain. Epigenetic changes caused by acute activation of HS signaling [22,23] may contribute to such long-term impacts.

## 5. Mechanisms of Cell-to-Cell Heterogeneity in HS Signaling Activation

The brain of mice prenatally exposed to environmental stress shows cell-to-cell heterogeneity in the nuclear localization of HSF1 [19]. Similarly, cell-to-cell heterogeneity in HSF1 aggregation in the nucleus, associated with the apoptotic phenotype, was shown in human cancer cells [24]. As mentioned above, research using HS reporter systems revealed that such cell-to-cell heterogeneous HS signaling activation is involved in tissue and organ pathology.

Recent studies have demonstrated that such heterogeneous and dynamic HSF1 activation is controlled by liquid–liquid phase separation to form HSF1 small nuclear condensates, as well as by HSF1 phosphorylation [25,26] (Figure 2). Another study suggested that heterogeneity in the HSF1 activity states is consolidated through interactions among cells [27] (Figure 2). On the other hand, stochastic activation and inactivation of promoters, so-called transcriptional bursting, was found in many genes, including *HSPs* [28] (Figure 2). Among these genes, the *HSP70* and other chaperone genes have particularly simple genomic structures (no introns, under the control of a strong promoter, etc.), which may contribute to their highly obvious heterogeneity in expression among cells. Immediate and heterogeneous changes in gene expression are anticipated to be required to diversify the adaptability of cells to respond to rapid environmental changes. As a regulatory mechanism of transcriptional bursting, AKT/ MAPK-mediated controls of transcription elongation in mouse embryonic stem cells were reported using CRISPR library screening [28].

In addition to transcriptional bursting, a recent paper reported a new mechanism, transcriptional amplification, in the regulation of *HSP70* and four other genes flanking the *HSP70* locus [29] (Figure 2). They found that an association with nuclear speckles—liquid-droplet-like nuclear bodies containing RNA-processing proteins, transcription factors, and RNAs—correlates with the several-fold boost in gene transcription when cells are exposed to environmental stress. This transcriptional amplification is distinct from transcriptional bursting, in which genes pulse on and off for extended periods of time [28,30]. 

These findings above suggest that the heterogeneity in the immediate transcription of molecular chaperones in a novel environment is driven by multi-folded mechanisms. They provide important clues to understanding the mechanisms of cell-to-cell heterogeneity in HS signaling. However, we have yet to see the whole picture that describes how these heterogeneous events interrelate and affect each other. 

## 6. Challenges and Perspectives

We discussed recent findings on the roles and mechanisms of HS signaling activation in cells exposed to cellular stress and the progress in the development and use of HS reporters. As demonstrated in previous work [20,21], site-specific FLPo-FRT genetic recombination systems have been invaluable tools for tracing heterogeneous cellular phenotypes associated with differential activation of HS signaling among cells. Similarly, Cre-loxP genetic recombination systems have been used for spatial and temporal controls of gene knockout and knock-in as well as cell lineage tracing in rodent models. A recent paper showcased the selective maternal or paternal germline recombination in several Cre-loxP mouse lines and warned about the possibility of the misinterpretation of the results obtained using these genetic recombination tools [31]. The authors proposed an optimal breeding strategy to avoid this issue in the mouse Cre-loxP recombination system. Considering similar breeding schemes may be important to minimize the risk of inconsistent recombination in using the mouse FLPo-FRT recombination system (Figure 1b).

A study using reporter transgenic mice revealed that the peak of *Hsp70*-promoter-driven HS reporter expression is delayed compared to the peak of *Hsp70* mRNA production [14], demonstrating a limitation of reporter systems in terms of time resolution. Another obvious limitation of the HS reporter system for detecting cellular response to various stressors is that it only provides information on the activation of a single signaling pathway. There are other important stress response pathways, such as autophagy, unfolded protein response (endoplasmic reticulum and mitochondrion), remodeled proteasome, and DNA damage response pathways [32,33]. Identifying their differences, commonalities, and interactions is essential to elucidating the whole picture of the cellular stress response.

A comprehensive understanding of the epigenetic effects of HS signaling activation is also important. To date, only a handful of papers have addressed this issue. Such epigenetic mechanisms can exhibit long-lasting effects even by brief activation of HS signaling during the transition to (or short exposure to) a new (or harmful) environment. The genetic reporters discussed in this review will serve as powerful tools combined with recent cutting-edge molecular tools for epigenomics and in vivo physiological imaging. 

## Figures and Tables

**Figure 1 cells-11-01561-f001:**
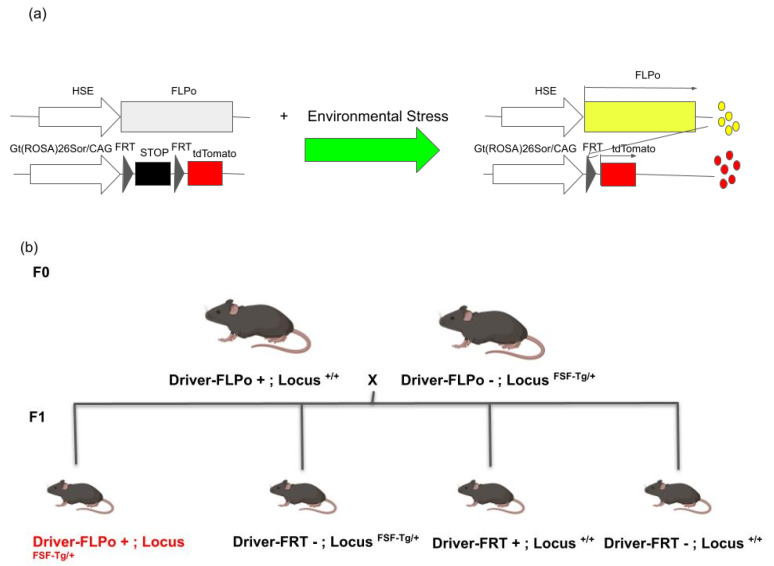
Approaches for lineage tracing of stressed cells in mouse models. (**a**) Schematic showing transgenic constructs of Flippase-FRT system by which stressed cells can be continuously monitored after Heat Shock signal activation. (**b**) Suggested mating strategy. To avoid germinal recombination, which can confound accurate cell lineage tracing, the recombinase-containing transgenic mouse line needs to be separately maintained until crossing with the reporter-transgene-containing mouse line for lineage tracing experiments. F0 and F1 are parent and offspring animals, respectively. The F1 animal highlighted in red can be used for the experiment. FSF-Tg: FRT-stop-FRT-Transgene.

**Figure 2 cells-11-01561-f002:**
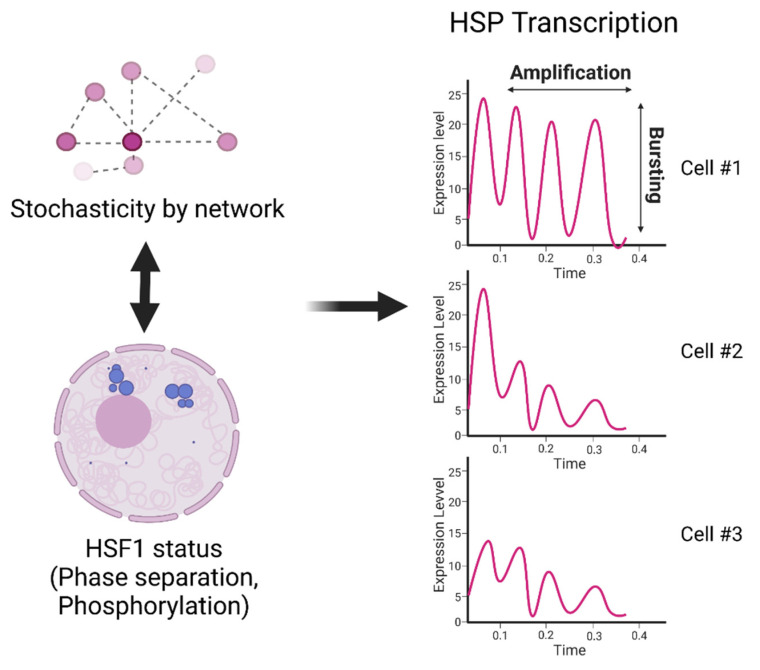
Hypothetical model of heterogeneous activation of HSP transcription. Stochastic cellular states can be generated by intercellular communications within the network. These communications and other mechanisms, such as liquid–liquid-phase separation at *HSP* gene loci, initiate heterogeneous HSF1 states between cells. These differences may cause differences in the dynamics of *HSP* transcription between cells through transcriptional bursting and amplification.

## Data Availability

Not applicable.

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
