# Peer review of "Tracking the Activation of Heat Shock Signaling in Cellular Protection and Damage"

_cells, 2022, doi:10.3390/cells11091561_

Round 1
Reviewer 1 Report
The title is still not summarizing the content of the review. The involvement in diseases is only marginally described.
Restructuring is still required because the red line of the Ms is not clear for me.
The title of the Ms is a subchapter title (5) why
The authors need to make clear what are the limitations in HS signaling and describe applications more detailed. "the activation of HS signalling also cahnges the epigenetic status ... "is too general.
Author Response
Thanks a lot for reviewing our manuscript and your comments. We have revised the manuscript accordingly.
The title is still not summarizing the content of the review. The involvement in diseases is only marginally described.
We have removed disease from title and changed it to influence the context better.
Restructuring is still required because the red line of the Ms is not clear for me.
We have restructured by moving paragraphs and revised manuscript extensively.
The title of the Ms is a subchapter title (5) why
We have changed title to reflect an overall claim instead.
The authors need to make clear what are the limitations in HS signaling and describe applications more detailed. "the activation of HS signalling also cahnges the epigenetic status ... "is too general.
We have extended the description to explain it in detail in the section 3 and 6.
Reviewer 2 Report
While the authors have made an effort to improve their manuscript, I'm afraid I still cannot support its publication. I willing to admit that I may have disproportionate expections as to the linguistic and logical qualities of a text. However, I find it impossible to give the contents of the text the credit that it might deserve when its linguistic quality makes it very difficult to read and understand. Although the new paragraph #5 may have fixed the key issue pointed out in my major comment, a number of the minor issues have not or only partially been corrected. And then of course, there is the generally poor writing style that lingers throughout the text. I suspect that the text could only be fixed by a very close interaction between the scientists (the authors) and a professional editing service.
Author Response
While the authors have made an effort to improve their manuscript, I'm afraid I still cannot support its publication. I willing to admit that I may have disproportionate expections as to the linguistic and logical qualities of a text. However, I find it impossible to give the contents of the text the credit that it might deserve when its linguistic quality makes it very difficult to read and understand. Although the new paragraph #5 may have fixed the key issue pointed out in my major comment, a number of the minor issues have not or only partially been corrected. And then of course, there is the generally poor writing style that lingers throughout the text. I suspect that the text could only be fixed by a very close interaction between the scientists (the authors) and a professional editing service.
We appreciate your suggestions. We fixed linguistic issues by extensive revisions by working with external support from professional editors at Yale university. We also improved logical quality by moving paragraphs to make manuscript read better.
Round 2
Reviewer 1 Report
The authors have addressed most aspects which were raised by the reviewers and teh ms has improved.
Author Response
Thanks a lot for reviewing our manuscript and your comments. We have revised the manuscript accordingly.
Reviewer 2 Report
- As I stated before, this commentary does address an interesting issue and the revised version (R1) does so reasonably well. However, the quality of writing is still not satisfactory. A few examples are given below, but it is not up to me to list those issues in detail. At this point, it is up to the editor of the journal to decide whether a relatively poorly written text should be accepted solely on the basis of its contents.
- Not a good start, 1st paragraph: the first sentence starts with a typo; homeostatis instead of homeostasis. the penultimate sentence "Upon exposure to such as heat" is incomplete.
- The manuscript still suffers from many of the issues I had raised when it was first submitted:
- Definite and indefinite articles are still incorrectly used in many many instances.
- The use of the case for proteins/genes is still inconsistent. Both HSF1 and Hsf1, or HSP and Hsp can be found without any obvious system. There is also Cre-loxp system, which should be LoxP or loxP.
- There are still instances of "expressions", a plural which just doesn't exist in this context.
Now that I have received the correct version of the revised manuscript, I understand the authors' brief response, and I do indeed come to a different conclusion. The quality of the text has made a quantum leap. It still isn't perfect (there are still instances of HSF1 and Hsf1, and HSP70 and Hsp70, and definite/indefinite articles are still missing in some cases and not always correct), but it is far more acceptable.
Author Response
Thanks a lot for reviewing our manuscript and your comments. We have revised the manuscript accordingly.
This manuscript is a resubmission of an earlier submission. The following is a list of the peer review reports and author responses from that submission.
Round 1
Reviewer 1 Report
The work could be very interesting since it aims to describe the more recent strategies used to trace the activation of the HS signaling in cells. However, in my opinion, both the form and the content need substantial revision.
Below are some examples of what I mean.
1. FORM
Punctuation
In my opinion, the punctuation should be improved to make the reading more fluent.
Abbreviation
In general, the meaning of each abbreviation should be clarified the first time it is used. For instance, in line 43 you write “HSE-YFP”, but what does HSE stand for?
Line 8: “..HS activation happens..” In line 7 you write that HS stands for Heat Shock. However, considering the use of this abbreviation along the text, it should mean Heat Shock signaling, or you should write signaling after HS along all the text, as you do in lines 12, 36...
Numerous times along the text you randomly use both uppercase and lowercase letters, or italicized letters to write the symbols of a protein, a gene, or a mRNA. There is a detailed nomenclature to indicate proteins or genes symbols, that distinguishes among, lower and upper case letters, italicized letters, and so on, and that is also species-specific. In my opinion, you should pay more attention to these guidelines, in order to make the manuscript more precise and accurate.
For instance, in line 39 you write Hsp, but above (lines 20,23,27, 35) you use the abbreviation HSPs (uppercase letters) to indicate the heat shock proteins.
In line 36, as well as below (lines 42, 53, 56, 86, 107), you write “HSP70 promoter…” Is this HSP70 referred to the gene? Why do you use both uppercase (lines 36, 42, 86, 107) and lowercase letters (lines 53, 56, 62)?
Again, in line 45 you write “activation of Hsp70” What does Hsp70 stand for? The protein or the gene? If it indicates the gene, you should write “activation of HSP70 transcription”
In lines 81, 85, and 92, you write Hsf1, but above, and also below you write HSF1 (uppercase letters).
And so on.
2. CONTENT
In my opinion, many sentences are not well written and are a little bit confusing, so it’s very difficult to understand the meaning. You should pay more attention to the terminology used, and to the sentence construction, in order to make the text more clear and easy to understand.
Lines 19-33. The introduction does not introduce the topic/topics that will be discussed in the following paragraphs. In my opinion, it should give some notions and anticipations about the arguments that are going to be discussed below. Otherwise, this could be a separate paragraph and not the introduction.
Lines 42, 43: “the HSP70 promoter was used to drive Yellow Fluorescence Protein (YFP) encoding gene” Maybe it would be better to write “was used to drive the transcription of the YFP gene”. The words used should be precise.
Line 43: “The integration...” What do you mean by the word “integration”? Maybe that the reporter system was integrated into the yeast genome. In my opinion, you should pay more attention to the words used and to their meaning.
Line 53: “…which includes the transgene of luciferase” What do you mean? Is the “transgene of luciferase” a type of transgenic mouse? This is what I can deduct from this sentence, so you should pay more attention to the construction of the phrase.
Reviewer 2 Report
The basic idea behind this commentary is very interesting and the topic is definitely worth it. The overwhelming majority of papers about the heat-shock response or about the role of Hsf1 in this are based on experiments with cell populations, rather than individual cells. But of course it is the latter that need to be protected by this adaptive response. Hence, it is absolutely necessary to understand what happens at the single-cell level and what the underlying mechanisms are that lead to cell-to-cell heterogeneity. The authors have previously published a number of papers on this issue and on novel tools to track this heterogeneity. Unfortunately, the manuscript suffers from a rather onesided presentation of what has been done by others, and from a relatively poor writing style.
Major comments:
- While the authors have indeed made substantial contributions, there are relevant contributions from others, too. A quick search of the web pulls out the following: Gaglia et al. (2020) Nature Cell Biology 22, 151; Guilbert et al. (2020) FEBS J. 287, 5345; other papers from the Pincus lab, other than the one(s) cited here, such as Zheng et al. (2018) Cell Rep. 12, 3099. Perhaps a comprehensive literature survey would identify yet others.
Minor comments:
- Definite article (e.g. the) versus indefinite article (e.g. a): Most sentences have issues with the proper use of "the" versus "a" versus none at all.
- First statements of the Introduction: This is not correct as stated. Proteostasis is the maintenance of protein homeostasis. It involves more than just the increased synthesis of HSPs. Preexisting molecular chaperones are critical, and there are also critical changes in small molecules (e.g. glycerol).
- Line 27: "transcription" doesn't have a plural.
- Line 30: the vast majority of "harmful chemicals" do not affect HSF1.
- Line 37: the promoter contains HSF1 binding elements, not domains. This also needs to be corrected further down in the manuscript.
- Line 38: How does the use of the "HS luciferase reporter" exclude effects of HSF2?
- Line 40: the article in the statement "The luciferase-based HS reporter...."could mislead readers to think that the very same reporter construct could be used in mammalian cells and in yeast.
- Line 44: "reorganization" ?
- Subheadings: it is very strange/unusual to use the past tense in subheadings. Presumably these statements are still valid today.
- Line 48: proteostasis is not "triggered", it is maintained or it breaks down.
- Line 53: transgenic mice "carry" or "have" a transgene, they don't include it.
- Line 63: in the sentence that starts there it is not clear whether both statements refer to mice or tissue culture cells.
- Line 66: not sure what an "intrinsic" mRNA is.
- Line 70: the c of this nematode should be spelled out here (this is its first mention) and capitalized.
- Line 81: Be consistent with the case used for things like HSF1 (here and later, too).
- Line 83: The sentence that starts here is grammatically incomplete.
- Line 85: I am assuming that reporter plasmid (of HSP70 promoter activation) and driver plasmid (expressing flippase) in this context are one and the same, but that's not clear.
- Line 96: there is no plural of "expression".
- Line 108: "there were random distribution"? It's either singular or plural, but not both.
- Line 142: The chaperones lost an e.
Reviewer 3 Report
The title is a bit cryptic and should be reivsed.
Keywords should not be numbered.
Introduction: at present 13 HSF are known for human cells. The introduction is very short and very general and should be extended and specified.
The Ms needs major restructuring. It is not clear whther it is a research article or a review or a commentray?
The Figure is not very helpful.